# Recent Advances and Challenges in Cancer Immunotherapy

**DOI:** 10.3390/cancers14163972

**Published:** 2022-08-17

**Authors:** Chelsea Peterson, Nathan Denlinger, Yiping Yang

**Affiliations:** Division of Hematology, The Ohio State University Comprehensive Cancer Center, Columbus, OH 43210, USA

**Keywords:** immunotherapy, immune checkpoint inhibitor, chimeric antigen receptor (CAR)-T cell therapy, neoantigen, cancer vaccine, tumor microenvironment (TME), tumor-associated macrophages (TAMs)

## Abstract

**Simple Summary:**

Immunotherapy helps a person’s immune system to target tumor cells. Recent advances in cancer immunotherapy, including immune checkpoint inhibition, chimeric antigen receptor T-cell therapy and cancer vaccination, have changed the landscape of cancer treatment. These approaches have had profound success in certain cancer types but still fail in the majority of cases. This review will cover both successes and current challenges in cancer immunotherapy, as well as recent advances in the field of basic tumor immunology that will allow us to overcome resistance to existing treatments.

**Abstract:**

Cancer immunotherapy has revolutionized the field of oncology in recent years. Harnessing the immune system to treat cancer has led to a large growth in the number of novel immunotherapeutic strategies, including immune checkpoint inhibition, chimeric antigen receptor T-cell therapy and cancer vaccination. In this review, we will discuss the current landscape of immuno-oncology research, with a focus on elements that influence immunotherapeutic outcomes. We will also highlight recent advances in basic aspects of tumor immunology, in particular, the role of the immunosuppressive cells within the tumor microenvironment in regulating antitumor immunity. Lastly, we will discuss how the understanding of basic tumor immunology can lead to the development of new immunotherapeutic strategies.

## 1. Introduction

Immunotherapy harnesses a patient’s immune system to target cancer and has resulted in novel therapeutic approaches and unprecedented clinical outcomes [1]. Although immunotherapeutic approaches have found success in a variety of cancer subtypes and clinical scenarios challenges still remain [2,3,4,5]. Thus, comprehensive knowledge of how these therapies function is essential to address these challenges [6,7]. Tumor–host immune system interactions are heterogenous and certain characteristics can predict immunotherapy responsiveness. The tumor microenvironment (TME), in particular, affects immunotherapeutic response and immune evasion [8,9]. A deep understanding of host–tumor interactions is critical to fostering the development of novel and more effective immunotherapies. In this review, we will summarize the current landscape of immune checkpoint inhibition (ICI), adoptive cellular therapy (CAR T-cell therapy) and cancer vaccination. We will also outline new standards of care and examine immuno-oncology trials, with a focus on factors that affect immunotherapeutic results. In particular, we will explore the TME and how it influences immunotherapy outcomes and how to potentially address host–tumor interactions to improve immunotherapy efficacy.

## 2. Current State of Cancer Immunotherapy

### 2.1. Immune Checkpoint Inhibitor Therapy

Immune checkpoint inhibitors (ICI) are one of the most promising types of cancer immunotherapy. Multiple drugs have received FDA approval for more than nine cancer types over the last decade [10]. ICI therapy is based on the premise that T cells contain evolutionarily conserved negative regulatory markers that act as “checkpoints” to regulate activation [9]. Early after activation, T cells upregulate inhibitory receptor cytotoxic T lymphocyte antigen 4 (CTLA4) and, later, programmed cell death 1 (PD-1), which then bind to co-stimulatory ligands B7-1, B7-2 and PD-L1 or PD-L2, respectively [11,12]. Ligands are presented by tumor cells, regulatory T cells (Tregs), myeloid cells and antigen-presenting cells (APCs), which dampen cytotoxic T-cell activation, resulting in immune suppression and tumor growth [13,14,15]. Upon treatment with immune checkpoint inhibitors, inhibition is released and cancer cells are targeted and destroyed by the primed and activated cytotoxic T cells [16,17]. ICI has resulted in successful treatment for a variety of recalcitrant cancers [18].

ICI has achieved marked success in patients with previously dismal outcomes treated with conventional cancer therapies, such as chemo, radiation and targeted therapy. Durable responses suggest long-lasting immunological memory can be seen, established in patients who respond to ICI. FDA-approved ICI therapies are outlined in Table 1.

Ipilimumab, a therapeutic anti-CTLA4 human IgG1 monoclonal antibody, was the first ICI approved by the FDA and has since been utilized for treatment in a number of diseases [19,20,21,22,23]. Though some ICIs have been successful, some have also failed. Another anti-CTLA antibody, Tremelimumab, is an IgG2 isotype that failed to meet its primary endpoint in late-stage trials for advanced metastatic melanoma [24,25]. Ipilimumab and tremelimumab appear to have different antibody-dependent cell-mediated cytotoxicity (ADCC) effects, though it is unclear why. These antibodies may mediate depletion of tumor-infiltrating regulatory T cells via ADCC, modulating their clinical outcome [26,27]. Pembrolizumab (pembro) and nivolumab, two human IgG4 anti-PD-1 checkpoint inhibitor antibodies, were the first PD-1-targeted, FDA-approved therapies for refractory and unresectable melanoma [23,28,29]. Pembro, in particular, has had marked success, demonstrating an increase in overall response rate (ORR) in NSCLC and in melanoma patients compared to standard-of-care chemo. In certain cases, for melanoma (nivolumab/pembro) and NSCLC (atezolizumab), ICI has become the first line of treatment over chemo [30,31,32,33]. Combination therapy with anti-CTLA4 + anti-PD-1 has demonstrated significantly improved ORR for metastatic melanoma at 58% [34]. However, increased toxicity to combined treatment is common and remains a challenge to address.

Increased PD-L1 expression has been found to be associated with improved responses to pembro and PD-L1 expression level is used as a biomarker to guide the indication for ICI in certain cancer subtypes [35,36,37]. The type of scoring system used, whether tumor-derived PD-L1, TME-derived PD-L1 or a combined score, is important to note.

Anti-PD-L1 antibody treatments have also been proven effective in multiple types of cancer. Atezolizumab was the first approved anti-PD-L1 ICI for the treatment of urothelial carcinoma, and avelumab and durvalumab are approved for multiple types of solid tumor cancers [18,23,33,38,39,40,41,42]. Tumor PD-L1 expression has also been used as a biomarker for treatment indication for anti-PD-L1 ICI.

Microsatellite instability in cancer results from defective DNA mismatch repair (dMMR) proteins. The presence of microsatellite instability at high levels (MSI-H) has been associated with markedly improved outcomes in multiple cancer subtypes, including ovarian and colorectal cancer (CRC) [43,44]. Dramatically improved outcomes seen with MSI-H and ICI resulted in pembro being the first treatment approved in the disease agnostic setting based on the presence of MSI-H, regardless of cancer subtype [45]. A recent clinical study with dostarlimab, an anti-PD-1 ICI, in dMMR rectal cancer recently reported 100% clinical ORR and no recurrence to date [46].

Though ICI results in dramatic results for responders, only 20–30% of patients achieve a clinical response to ICI. Thus, in order to address the majority of patients, a deep understanding of additional checkpoint pathways and how they may be targeted is critical. Table 1 outlines ongoing clinical trials to investigate novel ICI agents. T-cell immunoreceptor with Immunoglobulin (Ig) and ITIM domains (TIGIT) is an inhibitory molecule present on CD8+ and CD4+ T cells, Tregs and natural killer (NK) cells that regulates T-cell immunity via the CD226-PVR pathway. Approximately two dozen monoclonal antibodies targeting TIGIT have been developed as both single agents and to be used in conjunction with anti-PD-1 or anti-PD-L1 agents [47]. Tiragolumab, an anti-TIGIT IgG1 antibody, when combined with atezolizumab (Table 1) showed improved clinical efficacy in NSCLC vs. atezo plus placebo and is currently being investigated in multiple Phase III trials in solid tumors [48,49]. Other anti-TIGIT antibodies, vibostolimab and etigilimab, have also shown promising clinical activity in early phase studies [50,51]. V-domain Immunoglobulin T-cell activation suppressor (VISTA) is another potential target. VISTA maintains naive T-cell and myeloid cell quiescence through binding to P-selectin glycoprotein ligand-1 (PSGL-1). VISTA binding activity is greater in low-pH and hypoxic settings, such as the TME. VISTA suppresses T-cell activation and reprograms macrophages to an immunosuppressive phenotype in the TME [52,53]. Anti-VISTA antibodies are under development, including a pH-selective VISTA blocking antibody that suppressed tumor growth in a mouse model of colon cancer [54]. T-cell immunoglobulin and mucin-domain containing-3 (Tim-3) is another co-inhibitory molecule expressed on T cells and leukemic cells, such as AML. Multiple Tim-3 inhibitors are being studied and have showed promise in treating MDS and AML (Table 1). Anti-Tim-3 ICI functions to target Tim3-expressing AML cells and boosts antitumor T-cell activity via its ICI activity [55,56].

Although immune checkpoint blockade has achieved remarkable successes, there remain challenges. There is still a significant portion of patients that do not respond at all and adverse effects of ICI, particularly combination therapy, need to be addressed [9,25]. Further study of alternative checkpoint inhibitor pathways that allow tumor escape and the influence of the TME on ICI will also be critical to future success.

### 2.2. CAR T-Cell Therapy

Adoptive cellular therapy (ACT) traditionally referred to three different approaches: infusion of tumor-infiltrating lymphocytes (TIL), genetically modified T-cell receptor (TCR) therapies and chimeric antigen receptor (CAR)-modified T cells [57]. The use of other immune cell types, such as natural killer cells (NK), CAR-NK cells, as well as CAR-macrophages (CAR-M), is being studied, though no therapies have yet obtained FDA approval. The most successful ACT has been CAR T-cell therapy (CART), which now carries a multitude of FDA-approved indications and is being utilized as standard of care worldwide for a variety of hematologic malignancies (Table 2). Here, we focus on successes in CART, as well as challenges facing CART and future areas for research.

CAR T cells were originally described by Eshhar et al., whereby a murine single-chain variable fragment antibody domain (svFC) was linked with a CD3ζ signaling chain and then inserted onto a human T cell. These first-generation CARs allowed for major histocompatibility complex (HLA)-independent activation of T cells when presented with a specific target antigen recognized by the svFC [58]. In 2011, costimulatory domains (most commonly CD28 or 4-1BB) were added to the CAR construct, resulting in much improved CAR T-cell expansion, persistence and pre-clinical efficacy [59,60]. Ultimately, these “2nd generation” CAR constructs have displayed unprecedented clinical efficacy, particularly when targeted against CD19 expressing B-cell lymphomas and leukemias and against B-cell maturation antigen (BCMA)-expressing multiple myeloma.

CART has found particularly profound success in diffuse large B-cell lymphoma (DLBCL), the most common subtype of non-Hodgkin’s B-cell lymphoma (NHL). In relapsed or refractory (R/R) DLBCL, outcomes were dismal. Patients unfit for or who relapsed after autologous stem cell transplant traditionally had an overall response rate (ORR) to next line of therapy of 20–30%, with a median overall survival (OS) of approximately 6 months [61,62,63]. ZUMA-1 (axicabtagene ciloleucel or axi-cel) and JULIET (tisagenlecleucel or tisa-cel) were pivotal trials for an autologous anti-CD19 CART (CAR19) product for patients with R/R DLBCL. Autologous mononuclear cells were collected by apheresis, purified to select for T cells, engineered with an Anti-CD19 scFV, expanded and reinfused into the patient. Both axi-cel (CD28 costimulatory domain) and tisa-cel (4-1BB co-stimulatory domain) had substantial post-infusion expansion in the majority of patients [64,65]. CAR19 provided an unprecedented ORR of 54–82%, a complete response (CR) rate of 40–54% and a median OS measured in years [64,65]. Real-world trials demonstrated comparable results and most patients who achieved a CR and 30–40% of all patients appear to remain in durable remission five years or longer post-CAR infusion [66,67,68]. CAR19 is now considered standard of care for R/R aggressive NHL after 2 or more lines of therapy and recent studies support shifting CART to the second line (and for which axi-cel was recently FDA approved) [69,70].

CART has also found success in R/R Multiple Myeloma (MM) by targeting BCMA. Idecabtegene vicleucel was approved in 2021, based on a phase 2 study showing that heavily pre-treated, relapsed refractory patients achieved 73% ORR and 33% CR rate [71]. Ciltacabtagene autoleucel is a second anti-BCMA CART that did well in heavily pre-treated patients with R/R MM, with a 97% ORR, 67% CR rate and 77% durable response at 1 year [72]. A retrospective study analyzed a similar cohort of patients who received non-CART therapy and found significantly worse outcomes compared to those seen with CART [73]. Continued clinical trials will be required to determine the optimal timing or CART in MM.

Four CAR19 agents and two anti-BMCM CART agents have now been FDA approved (Table 2). CART has transformed hematologic cancer treatment, yet challenges remain. The majority of patients who receive CART ultimately fail therapy, whether due to primary progression or response then relapse [64,65,74]. In acute lymphoblastic leukemia (ALL), lasting remissions have been demonstrated in pediatrics, but most adult patients require an allogeneic stem cell transplant post CAR19 due to high rates of relapse [75,76]. In order to address these challenges, more research into the mechanisms of CART resistance is required. CART failure includes either primary resistance to CART or response and then relapse post CART; each characteristically results from a different mechanism, though mechanisms of failure may not be mutually exclusive. Causes of CART failure can be grouped into tumor or disease-intrinsic factors, CAR T-cell product-specific mechanisms and CAR T-cell/host interactions. Loss or mutation of the target antigen (e.g., the CD19 extracellular epitope on leukemic/lymphoma cells) results in tumor intrinsic CAR T-cell failure [77,78]. Further, 10–20% of ALL patients will relapse with CD19 (-) leukemia [75,78,79]. However, in one study of patients who achieved a CR and then relapsed, 68% relapsed with CD19 (-) disease (with 27% not evaluable and only 4.5% with CD19+ disease). In the pivotal ZUMA-1 trial, 3/11 (27%) patients who failed CART who had evaluable tissue had lost CD19 expression at time of progression [64]. Alternative antigen and multi-antigen treatments are being developed to address antigen loss. Shah et al. found that anti-CD22-directed CAR T-cell treatment achieved 70% CR rates in ALL after failure of CAR19 [80]. Clinical trials of anti-CD22 following CAR19 failure in NHL have also been successful [81]. Though studies are promising, antigen loss of the new target (ex. CD22 or CD20) is also a potential drawback. Thus, several clinical trials integrating multiple antigen targets (ex. CD19/22) on a single CAR are underway, including a tri-specific CART targeting CD19/20/22, soon to open here at the OSUCCC [82,83].

Poor expansion and function are caused by CAR T-cell product intrinsic deficiencies. Clinical performance relies on CART viability, transduction efficiency and phenotype. Inadequate products can be due to manufacturing error, poor culture conditions or low-quality donor T cells due to past therapy and/or high disease burden at the time of apheresis [84]. Both patient status and the manufacturing process must be optimized to address CART product intrinsic mechanisms of failure [80,82]. Studies evaluating cell product collection pre- and post-bridging therapy and how cytoreduction may affect product quality are required. Point-of-care manufacturing, novel culture conditions and varying culture length are innovative CAR T-cell manufacturing methodologies. Point-of-care manufacturing reduces vein-to-vein time and may improve therapeutic results [82,85]. Recent studies utilizing IL7 and IL15 for the expansion phase of CART culture (instead of IL-2 utilized FDA-approved products) and a shorter 8-day manufacturing period have shown an increase in T stem-cell-like memory populations and have been hypothesized to improve expansion and overall effectiveness [86]. Commercial production of 4-1BB CAR19 using a 2-day, expansion-free approach has demonstrated encouraging results [87]. This shorter method boosted stemness and proliferative potential by expanding CAR T cells in vivo (in human), as opposed to ex vivo [87,88].

Tumor, host and CAR T-cell interactions result from interplay of the TME with the infused CAR T-cell product. Immunosuppressive TME interactions with CAR T cells can reduce expansion and increase exhaustion, with host systemic inflammation and tumor burden contributing as well [89,90]. Retrospective studies quantifying risk factors for CAR T-cell failure have identified extranodal disease sites ≥2, increased C-reactive protein and lactate dehydrogenase (inflammatory markers) and high metabolic tumor volume at the time of treatment as predictive of outcomes. [91] Jain et al. identified increased protumoral tumor-associated macrophage (TAM) markers, increased PD-L1 expression in the TME and increased mononuclear-myeloid-derived suppressor cells (MDSC’s) in circulation as factors that correlated with poorer response to CAR19 [92]. In CART for solid tumors, a more robust TME and increased tumor heterogeneity are hypothesized to be the primary reason for the lack of efficacy [93]. Both novel bridging and conditioning regimens, as well as radiation therapy pre-CART, are being studied to abrogate TME-mediated CART inhibition [94,95].

In CLL, Bruton’s Tyrosine Kinase inhibitor (BTKi) use prior to leukapheresis has been shown to improve CAR T-cell expansion and function, decrease T-cell exhaustion and also mitigate toxicity [96,97]. BTKi augments CART function via TME modulation, including reduced PD-1 and CTLA-4 expression, inhibition of Tregs, downregulation of B-cell chemokines and disruption of tumor cell adhesion/homing [98]. Here, at the Ohio State University Comprehensive Cancer Center (OSUCCC), we have achieved relative success by utilizing ibrutinib through leukapheresis and bridging prior to CAR T-cell therapy in patients with Richter’s Syndrome (DLBCL with antecedent CLL). Five of nine patients treated achieved a CR and four of these remained disease free long term [99].

Increased PD-L1 expression by TAMs is associated with poorer outcomes in CART [92]. Zuma-6 investigated the use of anti-PD-L1 atezolizumab post axi-cel infusion. Though atezolizumab was determined to be safe, clinical outcomes were similar to giving axi-cel alone [100].

Another approach to address mechanisms of CART failure is the use of novel salvage agents. The focus is on restoring CART proliferation and function to recapture a clinical response after failure. In a study of pembro given post-CART failure, 3/12 patients responded to therapy, with only 1 CR. Though outcomes were poor with pembro compared to results seen with other salvage agents, the responders to CAR T cells re-expanded and were more functionally active [101,102,103]. Thus, this is proof of concept that CART activity and proliferation can be recaptured post-CART failure with salvate therapy. In a point-of-care manufactured CAR19 product, TIGIT expression post CART was found to be associated with poorer outcomes. Anti-TIGIT ICI rescued CART function in vitro and in vivo and may be a promising agent post-CART failure. Lenalidomide is an immunomodulatory agent that enhances CAR T-cells’ antitumor efficacy by altering tumor cell receptor expression, via modification of the TME landscape and via direct effects on CAR T cells [104,105]. Multiple abstracts and studies have revealed the benefit of lenalidomide containing salvage regimens post CAR T-cell failure and clinical trials to study this further are warranted [101]. Continued research with robust clinical studies investigating the timing and appropriate population to use post-CAR immunomodulatory agents to enhance CAR T-cell function are required.

CART efficacy has also remained minimal in solid tumors, with no FDA-approved agents as of yet. This is due to both a lack of efficacy and increased toxicity seen in clinical trials [106]. Solid tumors are highly heterogeneous and CART requires a ubiquitously expressed target on tumor, with relatively low expression on normal tissue [106]. Further, solid tumors contain a highly robust and immunosuppressive TME, which impairs trafficking to and function within the tumor [107].

Though CAR T-cell therapy has found unprecedented success in CD19 and BCMA-expressing hematologic malignancies, it has still not been successful in the majority of patients nor in the majority of other hematologic and solid malignancies. In order to address this, further basic research into novel CART cell engineering strategies to enhance tumor recognition, TME infiltration, and improve anti-cancer activity is required.

### 2.3. Cancer Vaccines

Cancer vaccines aim to generate and amplify pre-existing T-cell and immune responses by providing tumor-specific or tumor-associated antigens (TAA) for elimination by the immune system [108,109]. Cancer vaccines purport to create persistent, targeted anticancer immunity. Developing tumor-specific antigens for cancer vaccines is challenging. TAAs are antigens selectively expressed or overexpressed in malignancies but also expressed in normal tissues. Targeting TAAs may induce autoreactive immune responses, resulting in organ toxicity and autoimmunity. Neoantigens result from carcinogenesis-related gene mutations. Neoantigens, which are not found in normal tissues, can be displayed on target cell surfaces, identified by T cells, and are not influenced by tolerance. Recently, the availability and affordability of next-generation sequencing has led to large-scale identification and the establishment of a plethora of neoantigens as potential immune system targets [108,109]. Cancer vaccine platforms are divided into four categories: cell-based, viral-based, peptide-based and nucleic-acid-based vaccines [110]. Table 3 outlines both FDA-approved cancer vaccines and ongoing trials. Sipuleucel-T (Provenge) was the first FDA-approved cancer vaccine for metastatic castration-resistant prostate cancer [111]. It is designed to elicit a prostate-cancer-specific immune response against prostatic acid phosphatase, an overexpressing TAA [112]. Other cell-based vaccinations include antigen-loaded dendritic cell (DC) vaccines produced from tumor lysates or mRNA, TAA peptides, TAA-coding mRNA or neoantigens. DC vaccines have shown promising results in clinical trials as monotherapy, including resulting in improved OS. Additional adjuvants, such as chemotherapy, could improve vaccine efficacy by inducing the release of danger signals by tumor cells and enhancing the immune response [113,114,115,116].

Oncolytic virus immunotherapy attacks tumor cells and stimulates antitumor responses. Herpes simplex virus and the adenovirus are used as vectors for specific genes and tumor antigen expression. Their mass-production speed and extensive host cell tropism have benefited clinical research [110]. The best clinical progress is T-VEC (Imlygic), a first-generation recombinant herpes simplex virus vector, which is FDA approved for treatment of recurrent unresectable melanoma [117,118]. Additionally, clinical trials with vaccines using adenovirus vectors, carrying immune-stimulating genes or providing TAAs, have induced strong antitumor immunity and been successful in HER2+ breast cancer and BCG-unresponsive non-muscular-invasive bladder cancer [119,120]. Peptide-based vaccines, including chemical and biosynthetic formulations of expected or known tumor antigens, generate a strong immune response against the tumor antigen. Polypeptide vaccine, DSP-788, induces cancer-cell-specific cytotoxic lymphocyte (CTL) and T-helper cell responses in a variety of solid and hematological tumor environments (Table 3) [121].

Nucleic acid vaccines also generate a robust MHC-I-mediated CD8+ T-cell response [110]. They can stimulate humoral and cellular immunity and encode full-length tumor antigens, allowing APCs to cross present several epitopes. Several DNA vaccines utilized in the treatment of cervical cancer have shown encouraging clinical efficacy. mRNA vaccines can encode immunostimulants, TAAs and tumor neoantigens. Immunostimulant-encoded mRNAs, such as TriMix, induce tumor cell death and release tumor antigens or are coupled with multiple-TAA-encoded mRNAs that produce robust CD8+ T-cell responses to improve response rates in patients [122]. mRNA vaccines encoding neoantigens lead to customizable vaccinations. Melanoma patients treated with personalized, multi-peptide NeoVax revealed long-term persistence of neoantigen-specific T cells exhibiting a memory phenotype, multiple TCR clones with distinct functional avidities and evidence of tumor infiltration [123,124]. Similar studies have been recapitulated in glioblastoma patients with comparable antitumor responses [125,126].

For a variety of immunotherapies (ex. ICI), increased tumor mutational burden has been linked to more potent immune responses and improved efficacy of treatments [127,128,129]. Neoantigens show strong individual heterogeneity; specific mutations across different types of tumors create different quantities of neoantigens. Thus, it is likely this will drive the application of this therapy to be more personalized [109]. The complexity of applying a personalized approach to cancer vaccination, with considerations to each individual TME, and further neoantigen study are areas that will benefit from further basic immunologic research in order to inform the development of effective cancer vaccines.

## 3. Basic Research in Cancer Immunology

### 3.1. The Tumor Microenvironment

Tumor and host interaction shapes local and systemic immunity to promote tumor development and the immunosuppressive tumor microenvironment [130]. The TME is heterogenous and varies by patient, cancer subtype and stage. TME composition influences cancer immunotherapy patient responses [8]. To improve immunotherapy efficacy, it is critical to understand TME cellularity and functionally. Tumor cells drive TME formation by forming physical barriers, inhibiting immune cells and recruiting immunosuppressive cells. Tumor cells secrete immunosuppressive cytokines (e.g., TGF-b, IL-10, VEGF), drive expression of inhibitory receptors and ligands (e.g., PD-L1/2, CTLA-4) and reduce tumor-specific MHC-I antigens. Tumor cells can generate chronic, weak antigen signals that drive T-cell reprogramming into an unresponsive, transcriptionally “exhausted” state. They also deplete nutrients and accumulate waste products, such as lactate and kynurenine, which inhibit T cells and create a hostile environment for effector cells [5,8] (Figure 1, created with BioRender.com, accessed on 15 August 2022).

The TME’s immunosuppressive function depends on the recruitment of stromal cells and immune cells (especially myeloid cells) and their re-direction towards pro-tumoral functions. Tumor stroma, composed of non-immune cells, such as carcinoma-associated fibroblasts (CAFs), tumor-associated vascular endothelial cells (TAECs) and extracellular matrix (ECM) components, forms a physical and immunosuppressive barrier, which allows angiogenesis to occur and spread [131,132]. Tumor cells, myeloid cells and CAFs have a complementary interaction in the TME. CAFs constitute over 80% of cells in pancreatic and breast tumors; these tumors also exhibit increased myeloid cell infiltration [132,133,134]. CAFs convert T cells into inducible Tregs and limit T- and NK-cell function. They increase myeloid infiltration, produce tolerogenic dendritic cells (tDCs), activate immunosuppressive M2-phenotypic macrophages by secreting TGF-b/IL-10 and remove APCs by inhibiting signal activity. TAECs influence immune cell movement, tumor cell intravasation and extravasation via angiogenesis [131,132]. TAECs can serve as non-professional APCs because they lack CD80 and CD86 co-stimulatory expression, triggering antigen-experienced T-cell effector capabilities but not naïve T cells. As non-professional APCs, TAECs reduce antitumor immunity and promote tumor growth [132].

Tumor-infiltrating lymphocytes (TILs) are various subsets of lymphocytes that are recruited into the TME. Their function depends greatly on TME composition and interacting pathways. TILs include CD8+ CTLs, CD4+ T-helper cells, Tregs, innate lymphoid cells (ILCs), NK cells and natural killer T (NKT) cells. CD8:CD4 TIL ratios have been studied as biomarkers in metastatic melanoma, NSCLC and colon cancer, with certain ratios correlating with increased ICI response predictions, diagnosis and survival rates in patients [1,8,135,136]. Immunologically “hot” tumors have heavy TIL infiltration. “Cold” tumors lack TILs and are unable to recruit/activate immune cells [137]. Hot tumors with inflammatory gene signatures respond better to ICI, while cold tumors are resistant [138]. These inflammatory gene signatures could be used as a clinical prognostic tool to evaluate ongoing trials and to help explain hot vs. cold TME status [137]. Tumor cells may induce cold TME by exhausting or functionally suppressing lymphocytes that enter the TME. Anti-angiogenic treatment combined with ICI to create high endothelial venules (HEVs) in the TME may enhance activated T-cell infiltration [139]. This was demonstrated in breast, pancreatic and glioblastoma tumor models, which are cold tumors, leading to sensitization of the TME for ICI and CART therapies [139,140]. Other autophagy-inhibiting targets may inhibit tumor growth in melanoma and CRC while enhancing ICI efficacy [141]. Continued research on novel methods to transform immune-cold environments into immune-hot environments is ongoing.

T-cell exhaustion is an area that has been extensively researched. Exhausted T cells can be “pre-exhausted” or “terminally exhausted” with increasing expression of immune checkpoints, such as CTLA-4, PD-1, Tim-3, lymphocyte activation gene-3 (LAG-3) and TIGIT, denoting a more terminally exhausted state [8]. Poor self-renewal, disrupted metabolism and gradual effector function loss characterize these cells [5,131]. In preclinical melanoma, ovarian, breast and bladder cancer models, targeting PI3K/Akt or Wnt, which control CD8+ T-cell infiltration and cytotoxicity and selectively inhibit Treg proliferation, enhances antitumor immunity and promotes tumor regression. Combinational therapies to normalize or sensitize the TME may improve ICI treatment efficacy [142,143,144,145,146,147,148].

### 3.2. Tumor-Associated Macrophages and Other Immunosuppressive Myeloid Cells

TAMs constitute most of the non-tumor stromal mass in solid tumors and modulate tumor growth and immunosuppression within the TME [149]. TAMs are protumoral and have M2-like functions [150,151]. They promote tumor growth and metastasis via anti-inflammatory cytokine secretion and immunosuppressive immune cell interactions and recruitment [152,153,154]. Similar to TAMs, myeloid-derived suppressor cells (MDSCs) are pathologically activated, immature, potent immunosuppressive cells at various stages in differentiation [155,156]. They are subdivided into mononuclear MDSCs (M-MDSCs), morphologically similar to blood monocytes, and polymorphonuclear (PMN-MDSCs), which are morphologically similar to neutrophils [157]. Their recruitment from bone marrow to secondary lymphoid organs and TME by cancer-cell-secreted growth factors promotes overall protumorgenic activity by inducing NK and T-cell inhibition, allowing tumor immunoevasion [158]. Poor prognosis and OS were correlated with solid tumor MDSC abundance [159,160]. Tumor-associated neutrophils (TANs) also have critical functions within the TME. N1s instruct effector T cells to reject tumor cells and N2s dampen the immune system by enlisting M2s and Tregs [132]. Shorter survival rates in HCC and poor prognosis for DLBCL have both been associated with TANs [161,162]. Due to their abundance in blood and their immediate reaction to inflammation and injury, their influence in the TME can set the tone for other immunosuppressive cells. DCs also infiltrate the TME but quickly adapt regulatory or tolerogenic phenotypes that promote tumor growth and immunoevasion [163,164,165]. Conventional DC types 1 (cDC1) and 2 (cDC2) play a key role as APCs and activate T cells for antitumor responses [166,167,168]. cDC1s and cDC2s in the TME are associated with a good prognosis in various cancers, while plasmacytoid DCs (pDC) frequencies are associated with a worse PFS and OS [169,170,171,172,173].

TAM functions are influenced by the TME, cancer type and stage [174]. Petty et al. described the role of hedgehog (Hh) signaling in polarizing TAMs towards an M2 phenotype and regulating CD8+ T-cell-mediated antitumor immunity [175,176]. Tumor-derived sonic hedgehog (SHH), a Hh ligand, drives TAM M2 polarization resulting in downregulation of CXCL9 and CXCL10 signaling and suppression of CD8+ infiltration into the TME [175]. TAMs also inhibit CD8+ T-cell activation by depleting essential metabolites for proliferation [174]. They inhibit T-cell function by producing IL-10, TGF-b and PGE-2 and upregulating PD-L1 [174,176,177]. TAMs are a major source of elevated CCL2 expression in human glioblastoma, which correlates with reduced OS, through promotion of CCR2+ M-MDSC infiltration [178,179].

TAMs correlate with poor outcomes in many cancer subtypes [180]. TAMs’ role as major carriers of checkpoint inhibitor ligands (e.g., PDL-1/2, B7-H4, VISTA) and as mediators of T-cell exhaustion has been well described [181]. Studies have also revealed how an increased concentration of M2-polarized TAMs mediate anti-PD-L1 ICI resistance in HNSCC and in prostate cancer [182,183].

Given their role in modulating immunotherapy, TAMs represent a promising target to augment immunotherapy. Reducing TAM presence in the TME by depletion or preventing trafficking and reprogramming TAMs to an immune-activating, M1-like phenotype are approaches being studied. Trabectedin is an FDA-approved agent for soft tissue sarcoma which targets TAMs to inhibit CCL2 and IL-6 production. When used in combination with an anti-PD-1 ICI in an ICI-resistant pre-clinical sarcoma model, trabectedin allowed for the recapture of response to anti-PD-1 therapy [184]. Targeting of PI3k-γ in myeloid cells using eganelisib (IPI-549) has also shown promise in pre-clinical models by restoring sensitivity to ICI [185,186].

Studies evaluating the repolarization of TAMs to a more M1-like, T-cell-activating phenotype are ongoing. Activation of toll-like receptors (TLR) on macrophages can lead to M1 polarization. Local delivery of a TLR7/8 agonist, telratolimod, in combination with ICI in murine melanoma boosted both local and systemic antitumor immunity [187]. Another novel approach utilizes low-pathogenicity influenza aviruses (IAVs), which have both oncolytic activity and also result in significant repolarization of TAMs to an M1-like state. When IAVs were combined with a novel B7-H3 ICI, responses were dramatic in a resistant NSCLC model [188]. Hh signaling blockade with vismogedib reduced TAM M2 polarization, increased CD8+ T-cell infiltration and suppressed tumor growth in murine lung and HCC [175]. Synergy between Hh inhibition and ICI was demonstrated as well with greater reduction in tumorigenesis than seen in either agent alone [175]. CART (and, in particular, CART targeting solid tumors with a more robust TME) and ICI may benefit from innovative approaches to abrogate TAMs’ immunosuppressive activity in the TME.

### 3.3. Approaches to Enhance ICI Therapy

Given the response rates seen thus far with currently FDA-approved ICI therapies, it is unlikely that one type of ICI will overcome the various mechanisms of resistance employed by different types of tumors. Targeting aspects of the TME to overcome tumor resistance has shown promise in enhancing ICI therapy [189].

Targeting the CXCR4/CXCL12 (SDF-1) signaling pathway can help overcome the physical barrier of the TME and enhance ICI. CXCR4 is upregulated on MDSC/TAMs, playing a role in intratumoral fibrosis, and is associated with poor prognosis in several types of cancer. Nanocomplex technology, polymer-based combinatory approaches and liposomal formation have been used to combine anti-PD-L1 agents and CXCR4 antagonists to overcome ICI resistance [190,191,192]. Increased effector T-cell infiltration, decreased Treg and MDSC populations and inhibition of primary tumor growth and metastasis were all observed in several tumor models. Nanoparticle technology applying a CSF1R inhibitor in combination with ICI allowed for the development of a sustained codelivery method that successfully reprogramed TAMs to an antitumoral M1-like phenotype and enhanced their phagocytic capabilities in a melanoma model [193]. In glioblastoma, overcoming the TME physical barrier is also being explored. One group combined brain-tumor-targeted peptide-coated extracellular vesicles, loaded with small interfering RNA (siRNA) against PD-L1, and then delivered them with bursts of radiation therapy [194]. Dual inhibition of PI3K/mTOR pathway combined with the microtubule targeting chemotherapy, paclitaxel, along with ICI, induced sustainable DC, T cell and NK responses, both locally in the TME and systemically [195]. Similar results targeting cancer stem cells with the polyketide antibiotic, Mithraymcin-A, and ICI in a CRC mouse model resulted in turning an immunologically cold tumor hot by increasing CD8+ T-cell infiltration and decreasing quantities of MDSC/TAMs in the TME [196].

New ICI targets are currently being studied as well. NKG2A is a checkpoint inhibitor expressed on subsets of cytotoxic T cells and NK cells. It binds to the MHC-I molecule HLA-E (Qa-1b mouse homolog) and is regulated in a TAP1-dependent manner [197]. Co-deletion of TAP-1/Qa-1b or blocking NKG2A with monalizumab unleashed effector cell activity and reversed resistance to ICI, resulting in tumor control [198,199]. Another target previously mentioned, used as both a monotherapy and in combination with ICI, is VISTA. Preclinical studies using L557-0155, an inhibitor for VSIG-8 (a VISTA receptor) promoted cytokine production by T cells and suppressed melanoma growth [200]. An orally administered combinatory small molecular inhibitor of VISTA and PD-L1, CA-170, showed similar antitumor efficacy in a number of mouse tumor models, prompting advancement towards clinical trials [201].

Diverse techniques to overcome mechanisms of resistance to ICI, including the immunosuppressive TME, as well as alternative ICI pathways that allow for tumor escape are ongoing. Continued focus on turning these basic discoveries into clinically applicable therapies to augment ICI is required.

### 3.4. Strategies to Improve CART Therapy

CART efficacy has achieved a high degree of success in hematologic malignancies. However, certain patients with risk factors, such as tumor bulk and a highly immunosuppressive TME, have worse outcomes. CART in solid tumors, in particular, has been characterized by a lack of efficacy due to the highly immunosuppressive TME present, resulting in impaired CART trafficking and suppressed proliferation and activation within tumors [107].

The TME inhibits trafficking and CART by producing suppressive soluble factors and overexpressing negative immune checkpoints. Strategies to deplete immunosuppressive elements of the TME, such as TAMs, to negate these effects are being studies. TAMs and PD-L1 expression reduce CART expansion and result in increased exhaustion, according to multiple studies [90,92]. PF-04136309, a small-molecule inhibitor of the CCL2-CCR2 axis responsible for TAM recruitment, has been used in pancreatic adenocarcinoma with chemo to reduce TAM and Treg infiltration and increase CD4+ and CD8+ effector cell presence in the tumor stroma [202]. This agent has been proposed for use as part of novel conditioning regimens prior to CART to help reduce TAM-mediated suppression of CART expansion and function [202]. Blockage of colony-stimulating factor 1 receptor (CSF1R) signaling, involved in TAM recruitment, with pexidartinib (PLX3397), improved the efficacy of adoptive cell transfer in murine melanoma through inhibition of TAM recruitment and activation [203]. Folate receptor beta (FR) is highly expressed in M2-polarized protumoral TAMs. Anti-FR targeting CAR T-cells was utilized to condition melanoma and colon cancer prior to antigen-specific CART therapy and improved outcomes in CART-resistant murine models [204].

New CAR T-cell engineering strategies have also been used to boost tumor infiltration and anti-cancer activity in the hostile TME [205,206]. A CCL19/IL7 secreting anti-mesothelin CART product was used increase CART and non-CAR T-cell infiltration into the TME, resulting in growth inhibition of xenografted pancreatic cancer. An IL-15/IL-21 secreting CAR targeting GPC3 in HCC was used and found to help maintain TCF1 expression (critical for T-cell development, proliferation and memory formation in CART). This resulted in robust proliferation and expansion as well as superior tumor control and survival in mice [207]. One promising method is the use of TGF-beta knockouts to engineer CARTs more resistant to immunosuppression. A TGF-beta 2 receptor knockout CART, developed using the CRISPR/Cas9 system, was found to reduce Treg-induced conversion, prevent CART exhaustion, improve in vivo elimination of tumor and improve CART memory subset formation to improve long-term efficacy [208]. One novel CART product was engineered to express CCR8 to improve homing to the site of the tumor, as well as a dominant negative TGF-beta 2 receptor to shield CART from TME-derived immunosuppression. This resulted in increased and more sustained infiltration of CARTs in xenografted cancer models [209]. Continued work developing these novel engineered CARTs into clinical-grade products is required to determine what strategies may provide the best outcomes and provide more insight into what is efficacious in solid tumors.

### 3.5. Neoantigens in Cancer Vaccination

Neoantigen vaccinations elicit a more powerful and specialized immune response than self-derived TAA vaccines [108]. Next-generation sequencing (NGS) of tumor DNA has led to customized clinical recombinant vaccines [210,211]. A patient’s HLA alleles might be used to identify a TAA or neoantigen for vaccine development and Sahin et al. demonstrated NGS-derived neoantigens elicit strong CD4+ and CD8+ responses [127,212,213]. New study domains include choosing and predicting how a vaccination strategy would function and utilizing aspects of the TME to assist these efforts.

Although early neoantigen vaccine trials have shown promising results, there remain failures and eliciting stronger antitumor responses is a continued goal. Coupling the binding properties of MHC-I with MHC-II has also presented challenges. A single MHC-I neoantigen is insufficient for antitumor immunity in murine models; thus, individualized vaccines should comprise neoepitopes expected to bind to MHC-I and MHC-II alleles [214,215]. MHC-II molecules have increased diversity and their open binding pockets make it difficult to predict a suitable binding motif [216]. Deep learning approaches that utilize artificial neural networks, inspired by biological neural networks, to predict ligand binding epitopes of MHC molecules are currently being investigated [217,218,219,220]. MARIA, a deep learning model, was used to analyze T-cell response data from a melanoma neoantigen vaccine study. This study demonstrated neoantigen candidates with high projected MARIA scores produced a CD4+ T-cell response post vaccination [221]. New AI-based applications and increased processing power help curb the complexity in predicting clinical outcomes and can begin closing knowledge gaps [215].

Although vaccines used as monotherapy show some potency, it is hypothesized that the vaccine-activated T cells are still suppressed by the TME. Combination approaches have been investigated clinically with current vaccines and also led to new insights in preclinical studies. Vaccines can reshape the TME, improving their effectiveness. Both combination and monotherapies used to enhance specific immune facets within the TME have shown potential to enable cytotoxic effects of vaccines in preclinical studies. Strategies, such as directly targeting TME vasculature (angiogenesis), targeting CAFs, and persistent cytokines, such as GM-CSF, IDO1, BAFF and other interleukins, with cancer vaccines have shown relatively consistent antitumoral effectiveness in preclinical models [222,223,224,225,226,227,228,229]. DNA vaccines utilizing novel nanobiomaterials include minimally invasive, injectable smart hydrogels [230]. These are scaffold-based cancer vaccines that allow for spatial and temporal control of antigen and other therapeutic agents and have shown strong antitumor effects via increased DC infiltration [231]. Targeting TAM infiltration, activity and polarization is also essential in the delivery of cancer vaccines. Combinatorial techniques have utilized biomimetic recombinant bacterial and viral vectors, nanoparticle and nanoemulsion technology for M1 agonists [232,233,234,235,236,237,238,239,240,241,242]. These have shown remarkable therapeutic efficacy through recruitment of lymphocytes, phenotypic transformation of macrophages from M2 to M1 and restoration of exhausted T lymphocytes, enhancing their ability to kill cancer cells in tumor models. With a deeper knowledge of bidirectional communication within the TME, future research will examine cancer vaccines that target immune pathways in the TME to boost vaccination efficacy. Neoantigen identification has developed greatly in the past decade, attributable to NGS, increasing computer capacity and better algorithms. Clinical studies that examine neoantigens as single or combinatorial immunotherapy targets are required to enhance this promising technology.

## 4. Conclusions and Future Directions

Over the last decade, cancer immunotherapy has markedly changed how we treat cancer patients. Immunotherapeutic modalities have found great success in a wide variety of settings and in patients with previously refractory disease. However, challenges remain. Further study of alternative checkpoint inhibitor pathways that allow for tumor escape and understanding the TME’s suppressive effect on ICI are critical areas of study required to develop more successful ICI therapies. CART has found good success in subsets of patients with hematologic malignancies but not for patients with a highly immunosuppressive TME nor with solid tumors. Targeting the immunosuppressive elements of the TME, including TAMs, has shown promise in improving both ICI and CART efficacy. Clinical studies utilizing these agents, either as pre-conditioning or in combination with ICI and CART, will be required to determine which approaches show the most promise. Targeting elements of alternative pathways with ICI has shown promise and will likely pave the way for new combination therapies. Novel engineering of CART to allow for improved CART trafficking and reduced immunosuppression within solid tumors may finally allow for improved outcomes to be obtained. Cancer vaccines have made great strides as well in recent years. In particular, studies utilizing neoantigen vaccines in combination with TME targeting have the potential to open up this exciting field.

Cancer immunotherapy is a modality that is rapidly becoming critical to the treatment of the majority of cancers. A continued focus on basic immuno-oncology research will drive our ability to develop novel therapies in the field and continue to drive successes on top of those we have achieved with cancer immunotherapy to date.

## Figures and Tables

**Figure 1 cancers-14-03972-f001:**
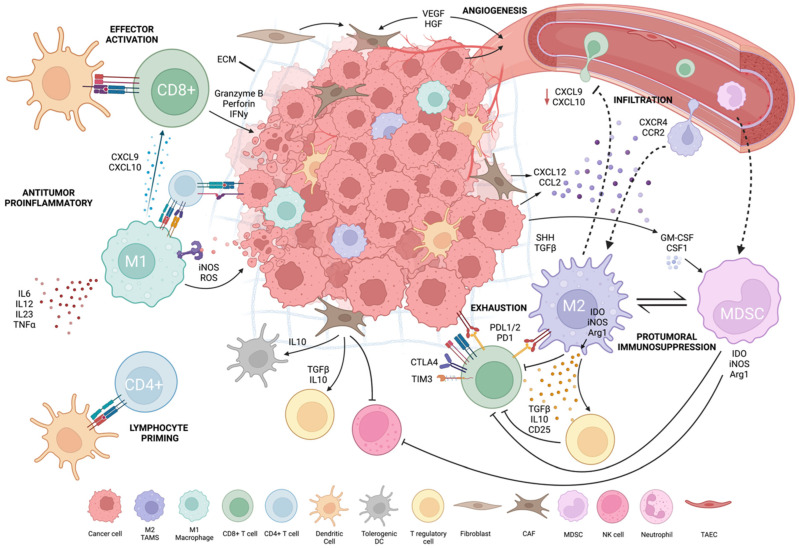
Tumor microenvironment.

**Table 1 cancers-14-03972-t001:** FDA approved, combination and ongoing clinical trials with ICI.

1.1 FDA Approved				
Approved Diseases		Agent	Target	First Approved
Melanoma, MSI-H/dMMR CRC, RCC		Ipilimumab		CTLA-4
Melanoma, NSCLC, HNSCC, HCC, cHL, MCC, MSI-H/dMMR CRC, DLBCL, urothelial cancer		Pembrolizumab		PD-1
Melanoma, NSCLC, HNSCC, HCC, MSI-H/dMMR CRC, cHL, RCC, urothelial cancer		Nivolumab		
NSCLC, urothelial cancer		Atezolizumab		PD-L1
MCC, RCC, urothelial cancer		Avelumab		
NSCLC, urothelial cancer		Durvalumab		
NSCLC, MPM, RCC		Ipilimumab + Nivolumab		CTLA-4 + PD-1
**1.2 Ongoing Trials**				
**Trial Identifier**	**Disease**	**Candidate(s)**	**Target**	**Phase**
NCT04165772	dMMR rectal cancer, dMMR ST	Dostarlimab	PD-1	II
NCT03563716	NSCLC	Tiragolumab, Atezolizumab	TIGIT	II
NCT02964013		Vibostolimab, Pembrolizumab		I
NCT03119428		Etigilimab, Nivolumab		I
NCT05082610	NSCLC, TNBC, ST	HMBD-002, Pembrolizumab	VISTA	I
NCT02608268	Advanced/metastatic ST	Sabatolimab (MBG453) ± PDR001	Tim3	I-Ib/II
NCT03066648	MDS, AML			Ib

Microsatellite instability-high (MSI-H), mismatch repair deficient (dMMR), colorectal cancer (CRC), renal cell carcinoma (RCC), Non-small cell lung cancer (NSCLC), head and neck squamous cell carcinomas (HNSCC), hepatocellular carcinoma (HCC), Classic Hodgkin lymphoma (cHL), diffuse large B-cell lymphoma (DLBCL), Merkel cell carcinoma (MCC), malignant pleural mesothelioma (MPM), triple negative breast cancer (TNBC), solid tumor (ST), myelodysplastic syndrome (MDS), acute myeloid leukemia (AML).

**Table 2 cancers-14-03972-t002:** FDA-approved and ongoing clinical trials with CAR T.

2.1 FDA Approved				
Trial Identifier	Approved Diseases	Agent	Target	First Approved
ZUMA-1 NCT02348216	DLBCL	Axicabtagene ciloleucel (Yescarta)	CD19	2017
ZUMA-5 NCT03105336	FL			2021
ZUMA-2 NCT02601313	R/R MCL	Brexucabtagene autoleucel (Tecartus)	CD19	2020
KarMMa NCT03361748	R/R MM	Idecabtagene vicleucel (Abecma)	BCMA	2021
TRANSCEND NCT02631044	NHL	Lisocabtagene maraleucel (Breyanzi)	CD19	2021
ELIANA NCT02435849	ALL	Tisagenlecleucel (Kymriah)	CD19	2017
JULIET NCT02445248	DLBCL			2018
CARTITUDE-1 NCT03548207	R/R MM	Ciltacabtagene autoleucel (Carvykti)	BCMA	2022
**2.2 Ongoing Trials**				
**Trial Identifier**	**Disease**	**Candidate(s)**	**Target**	**Phase**
NCT02315612	ALL, NHL	CD22-CAR	CD22	Phase I
NCT03019055	NHL, CLL/SLL	CAR-20/19-T cells	CD20/CD19	Phase I
NCT03960840	CLL/SLL, NHL, ALL	YTB323	CD19	Phase I
NCT05418088	CLL/SLL, CML with lymphoid blast crisis, ALL, NHL	CD19/20, CD22 DuoCAR	CD19/20/22	Phase I/II

Relapsed/refractory (R/R), diffuse large B-cell lymphoma (DLBCL), follicular lymphoma (FL), acute myeloid leukemia (AML), acute lymphoblastic leukemia (ALL), B-cell non-Hodgkin lymphoma (NHL), Mantle cell lymphoma (MCL), large cell lymphoma (LCL), chronic lymphocytic lymphoma (CLL), small lymphocytic lymphoma (SLL).

**Table 3 cancers-14-03972-t003:** FDA approved and examples of ongoing clinical trials for cancer vaccines.

3.1 FDA Approved				
Approved Diseases	Agent	Target/Function	Type of Vaccine	First Approved
Prostate cancer	Sipuleucel-T (Provenge)	Prostatic acid phosphate (PAP)	Cell: DC	2010 NCT00065442
Melanoma	Talimogene laherparepvec (T-VEC or Imlygic)	Replicate within tumors and produce GM-CSF	Oncolytic virus; Herpes	2015 NCT00769704
**3.2 Ongoing Trials**				
**Trial Identifier**	**Disease**	**Candidate(s)**	**Type of Vaccine**	**Phase**
NCT02301611	Melanoma	TLPLDC	DC	II
NCT00045968	Glioblastoma	DCVax-L	DC	III
NCT03632941	Breast cancer	VRP-HER2 ± Pembrolizumab	Adenovirus	II
NCT02773849	NMIBC	Nadofaragene firadenovec (Instiladrin)	Adenovirus	III
NCT04747002	AML	DSP-7888	Peptide	II
NCT03721978	Cervical cancer (cervical HSIL)	VGX-3100	DNA	III
NCT03444376	Cervical cancer	GX-188E	DNA	II
NCT03739931	TNBC, HNSCC, NHL, urothelial cancer, melanoma, NSCLC	mRNA-2752	mRNA	I
NCT01970358	Melanoma	NeoVax (Poly-ICLC (Hiltonol) + Neoantigen peptides)	Peptide	I
NCT02149225	Glioblastoma	GAPVAC	Peptide	I
NCT02287428	Glioblastoma	PNACV ± RT ± Pembrolizumab	Peptide	I

Non-muscle invasive bladder cancer (NMIBC), high grade squamous intraepithelial lesion (HSIL), triple negative breast cancer (TNBC), head and neck squamous cell carcinoma (HNSCC), non-Hodgkin’s lymphoma (NHL), non-small cell lung cancer (NSCLC), Glioma Actively Personalized Vaccine Consortium (GAPVAC), Personalized Neoantigen Cancer Vaccine (PNACV), radiation therapy (RT).

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
