# Peer review of "Recent Advances and Challenges in Cancer Immunotherapy"

_cancers, 2022, doi:10.3390/cancers14163972_

Round 1
Reviewer 1 Report
Authors described about cancer immunotherapy. The manuscript is well written.
1. Please, correct following: tumor derived PDL-1, TME derived PD-L1.
2. Please, mention tumor associated neutrophil.
Author Response
- Authors described about cancer immunotherapy. The manuscript is well written.
- Author response: Thank you!
- Comment about PD-L1:
- Author response: Thank you for pointing this inconsistency out. We have corrected the following: tumor derived PD-L1, TME derived PD-L1, line 90
- Suggestion to mention tumor associated neutrophils
- Author response: Tumor associated neutrophil (TAN) short description added on lines; 428-435.
Reviewer 2 Report
In this review, the authors provide an overview of the current status of ICI, CAR T cell, and cancer vaccine therapies. They discuss the tumor microenvironment and in particular the immunosuppressive role of macrophages and other myeloid cells. Finally they describe approaches to increase the efficacy of immune checkpoint inhibitory therapy, CAR T cells, neoantigens in cancer vaccines, and the influence of the microbiota on ICI and cancer neoantigens.
The topics addressed are many and it is difficult to follow the logic with which they are presented. It would be preferable to address fewer topics in a more thorough and articulate manner.
The microbiota for example is a topic of great interest that would deserve a much broader discussion. I would advise the authors to select fewer topics and articulate them in more depth.
Author Response
- In this review, the authors provide an overview of the current status of ICI, CAR T cell, and cancer vaccine therapies. They discuss the tumor microenvironment and in particular the immunosuppressive role of macrophages and other myeloid cells. Finally they describe approaches to increase the efficacy of immune checkpoint inhibitory therapy, CAR T cells, neoantigens in cancer vaccines, and the influence of the microbiota on ICI and cancer neoantigens.
- Author response: Thank you for your evaluation of this review.
- The topics addressed are many and it is difficult to follow the logic with which they are presented. It would be preferable to address fewer topics in a more thorough and articulate manner.
- Author response: As discussed below, we have removed the microbiota topic in order to narrow the focus on the three current pillars of cancer immunotherapy, ICI, CART and cancer vaccines. Our goal is to provide a comprehensive review of current therapies, as well as basic science advances, that may inform new avenues to perform these therapies.
- The microbiota for example is a topic of great interest that would deserve a much broader discussion. I would advise the authors to select fewer topics and articulate them in more depth.
- Author response: The microbiota section has been removed to allow the readers to focus more in-depth on the remaining topics covered in the review.
Reviewer 3 Report
The authors introduced well the recent advances and challenge in immunotherapies, giving a focus on the therapies FDA approved, the ongoing trials and the ICIs combination. The TME paragraphs are well described and also the figure is well depicted. Also the other paragraphs of the manuscritp go in detail focusing of the strategies to improve ICI and CAR T therapy, the use of neoantigen in cancer vaccination, the influence of microbiome un the treatment.
Here some corrections and personal considerations:
Line 215-216: "ailure" should be corrected with failure?
Line 237-239: Please the authors should revise this sentence.
The authors are focused their attention on a recent and updated argument and I think that they have described well the recent advances and challenge in immunothrapies, giving a short but detailed introduction also on Microbiome and ICIs and vaccines. Also the conclusions reflect all the manuscript point out the changes and improvment having during these years and the success of immunotherapies, but also focusing of the possibility in the future to continue to work hard in the immunology and oncology reasearch to develop new strategies to cure patients.
Author Response
- The authors introduced well the recent advances and challenge in immunotherapies, giving a focus on the therapies FDA approved, the ongoing trials and the ICIs combination. The TME paragraphs are well described and also the figure is well depicted. Also the other paragraphs of the manuscript go in detail focusing of the strategies to improve the ICI and CAR T therapy, the use of neoantigen in cancer vaccination, the influence of microbiome in the treatment.
- Author response: Thank you!
- Here are some corrections and personal considerations:
- Line 215-216: “ailure” should be corrected with failure?
- Author response: Line 215-216: Has been corrected
- Line 237-239: Please the authors should revise this sentence.
- Author response: Thank you for pointing this out, the sentence has been altered for better readability on line 244.
- The authors are focused their attention on a recent and updated argument and I think that they have described well the recent advances and challenges in immunotherapies, giving a short but detailed introduction also on Microbiome and ICIs and vaccines. Also the conclusions reflect all the manuscript point out the changes and improvement having during these years and the success of immunotherapies, but also focusing on the possibility in the future to continue to work hard in the immunology and oncology research to develop new strategies to cure patients.
- Author response: Thank you for your comments!
- Line 215-216: “ailure” should be corrected with failure?